# Analyzing the Structure of Closed-Loop Supply Chains: A Game Theory Perspective

**Ehsan Shekarian** [1,2,*] and **Simme Douwe Flapper** [2]

1   Department of Industrial Engineering and Management, University of Oulu, FI-90014 Oulu, Finland
2   Department of Industrial Engineering and Innovation Sciences, Eindhoven University of Technology, 5612 AE Eindhoven, The Netherlands; s.d.p.flapper@tue.nl
*   Correspondence: ehsan.shekarian@oulu.fi or e.shekarian@tue.nl or eshokryan@gmail.com

**Abstract:** Closed-loop supply chains (CLSCs) are seen as one of the circular economy's leading approaches for reducing our natural environment load. Many CLSC models require collaboration among different parties. Game theory (GT) offers a way to consider the profits of all parties in a CLSC, providing insight into the costs and benefits to the involved parties in an objective and quantitative way. Presently, available reviews on the use of GT, in the context of CLSC, are quite limited and consider only a few relevant elements. Here, we present a new and more extensive framework, focusing on the collaboration structure of CLSCs. It contains a content-based analysis of 230 papers based on a four-step systematic literature review process. The characteristics studied are channels for collection, reprocessing and selling, the planning horizon, and the types of games. The structures found are graphically reviewed, leading to 196 different structures. The results show that, so far, most attention has been paid to the dual-channel collection, where collection by two retailers (dual-retailer) is the most studied case. With respect to selling, most attention has been paid to situations with two selling channels (dual-selling), i.e., one channel managed by a manufacturer and one channel managed by a remanufacturer. Studies have prioritized the role of manufacturers as that of the leader and collector. Finally, a number of directions for further research are pointed out.

**Keywords:** closed-loop supply chain; game theory; structure; network; reprocessing; planning horizon

## 1. Introduction

According to the BBMG Conscious Consumer Report (www.bbmg.com), 51% of Americans are willing to pay more for environmentally friendly products and generally believe they can make a difference through their purchases from socially responsible companies. Therefore, change and modifications to the traditional supply chain management (SCM) structure have been an unavoidable phenomenon to make it consistent with the environment and consumers' expectations [1,2]. In this regard, closed-loop supply chains (CLSCs) can play an essential role by saving resources and reducing environmental pollution, which is beneficial for developing a sustainable economy [3]. The CLSC literature results show that it can be worthwhile for companies to collect products and adopt an active return approach [4]. Closing the loop is an approach to connecting the end point of a forward supply chain to its starting point through a reverse process to create a unique system and value. Forward activities include the usual actions from producing a new product to marketing. However, reverse logistics (RL) refers to all those actions needed to close the loop, such as product acquisition, refurbishing, recycling, remanufacturing, and remarketing [5]. The structure and design of an effective CLSC have essential ramifications for firms, regulatory bodies, and the market.

In a supply chain system, usually, the results of one decision-maker also depend on another decision-maker's choices, while everyone has their own preferences. An appropriate technique for studying such behavior by supply chain parties is game theory (GT) analysis. Since the introduction of GT by John von Neumann and Oskar Morgenstern in

their pioneering book "The Theory of Games and Economic Behavior", interest in potential applications of GT in SCM has peaked. GT is a mathematical theory suited to deal with interactive decisions in different situations. For example, cooperative and non-cooperative games help to design a supply chain by selecting an optimal coalition of partners and determining equilibrium points in trade conditions. There is significant literature on this relationship, especially based on GT. Thus, we are confronted with the following important question: What is the contribution of GT to CLSC models? In this regard, studying the collection process, which is the main element of a CLSC, is quite complicated, and an appropriate collecting method can increase the profits of the collectors. Many enterprises, such as Hewlett-Packard, Kodak, and Xerox Corporation have realized that they can save hundreds of millions of dollars each year by collecting the cores and reusing the post-consumer waste, facilitating substantial reductions in manufacturing costs as well as elevating their social prestige [6,7]. Regardless of these benefits, laws and directives such as the European Commission's Waste Electrical and Electronic Equipment (WEEE) Directive and Japan's Specified Household Appliances Recycling Law hold the entities responsible for the collection and proper treatment of end-of-use (EoU) products, aiming at reducing waste in landfills and incentivizing environmentally friendly product designs [8]. Many studies have focused on choosing the most effective channel for collecting used products, when there is a game among the entities. How to deal with the collected items is also important. There is a mutual interaction between the companies' after collection strategies and the structure of the created loop. This leads to considering different reprocessing options such as remanufacturing, recycling, and refurbishing that can affect the structure. Therefore, a critical question is, how to design the structure of CLSC channels based on GT?

Three reviews have studied the design of reverse and closed-loop supply chains based on GT. Guo, et al. [9] reviewed 62 papers between 2006 and 2016 on supply chain contracts focusing on RL systems and classified the literature concerning the supply chain structure and channel leaderships. De Giovanni and Zaccour [10] selected 73 papers between 2011 and 2018 and surveyed two critical issues in CLSC research, i.e., return functions and coordination mechanisms, and pointed out some research avenues. Recently, Shekarian [11] categorized the factors influencing CLSCs based on a review of 215 works published between 2004 and 2018. A careful analysis of the above reviews shows that the literature still lacks a comprehensive updated study that investigates CLSC networks using game-based models.

The contributions of this paper as compared with previous reviews are the following: (i) We investigate the role of CLSC participants from different points of view such as leadership, collection, and reprocessing, including remanufacturing, recycling, and refurbishing. (ii) We analyze the structure of CLSC channels generally categorized as single, dual, and multiple. For each of the three main activities (collection, reprocessing, and selling), more than one party may be included in a CLSC. This has consequences for all other parties involved in the CLSC. For this reason, unlike the other reviews, we pay explicit attention to this. (iii) We classify the CLSC models based on the different types of games used in the models. There are many different types of games that can be played. To make this clear, we explicitly indicate the type of game used. (iv) We categorize the CLSCs regarding the planning horizon. It is possible to deal with the time horizon of the models when considering GT. For example, a product, which is produced in the first period, will be remanufactured in the second period. These changes can be studied by applying two-period games. In this regard, we categorized the studied models into a few classes (i.e., single-period, two-period, multi-period (discrete), infinite planning horizons, and dynamic (continuous)). (v) We create graphical representations of the CLSC structures. "A picture tells more than a thousand words." To quickly get an overview of the different parties involved in a CLSC and their functions and relationships, we create graphical representations of all the different CLSC structures found. These small graphs are helpful for discussions with people in practice and help to quickly determine what is still missing in the academic literature. For a company to decide which role(s) it wants to play in the

context of a CLSC, it is important to know the consequences. In each CLSC, we indicate the main activity for the party who executed this activity.

The remainder of this paper is structured as follows: In the next section, we review the previous works; the applied research methodology, based on a systematic literature review, is explained in Section 3; in Section 4, we present the results; the results are discussed in Section 5; the conclusions and key findings of the paper are given in Section 6; and future research directions are indicated in the final section.

## 2. Relevant Works

### 2.1. Background on Game Theory (GT)

The first seminal paper discussing collection channels and designing a game on a CLSC is by Savaskan, et al. [12], which is the most cited paper in this field. After their work, many CLSC models were developed based on GT. As a fascinating part of microeconomics and macroeconomics, the use of GT in the problem of CLSCs assists decision-makers. It simultaneously presents a producer's benefits, by choosing environmentally friendly strategies for a product and the consumer, such as reuse and high return [13]. When there is cooperation or competition among the players of a loop to implement different duties from selling a new item to reprocessing activities such as remanufacturing, GT has proven be a useful tool for establishing an appropriate decision. For example, large collectors such as SIMS Metal Management and IBM's Global Asset Recovery Services collect products that are no longer desired by their owners. They play the role of the leader who coordinates the CLSC when there is competition among the parties. At this point, GT can be employed to analyze the effect of these leaders. Channel power structure affects the relationships among the players of the loop and the performance of the CLSC and parameters such as return rate [14]. Therefore, collection mode and power structure are significant challenges for CLSC management and affect both the customer and the environment [15].

### 2.2. A Critical Review

Different review papers have been published in the literature that have analyzed CLSCs from different aspects. In order to show the requisiteness, contribution, and importance of the present review among those in the literature, the reviews published after 2015 are briefly summarized in Table 1. The scope of the reviews varies regarding networks, value creation, inventory models, remanufacturing, quality, reliability, consumer behavior, WEEE, and financial approach. The studies have generally tried to classify CLSCs into different areas and have suggested areas of focus for future research. In this review paper, we investigated the content of 230 papers. The content-based method is prevalent in the sustainability literature [16].

Moreover, there are reviews before 2015 that investigated more than 100 papers in different areas such as production and operations management [17,18], environmentally conscious manufacturing and product recovery [19], inventory and production planning [20], contracts [21], and process and application in the industry [22,23]. There are other publications in the form of tutorials, critical reviews, and bibliometric studies that have targeted specific areas such as business economics of product reuse [24], network design [25], business perspective [26], strategic and tactical decisions [27], remanufacturing and recycling [28], and the whole area in RL and CLSCs [29]. However, none of them mentioned GT explicitly.

Other reviews surveyed CLSC models with a narrower focus and smaller size. CLSCs were classified by Wei, et al. [30] regarding core acquisition management. Cannella, et al. [31] studied 50 papers and investigated the inventory and order flow dynamics in CLSCs. Glock [32] investigated decision support models for returnable transport items by analyzing 33 papers. Focusing on papers published in the *Journal of Cleaner Production*, Govindan and Soleimani [33] surveyed 83 papers, up to 2014, related to RL and CLSCs. The bullwhip effect and complexity of CLSCs were reviewed by Braz, et al. [34] and Coenen, et al. [35] studied 56 and 64 papers, respectively. Malladi and Sowlati [36] reviewed CLSCs by study-

ing inventory routing problems with a focus on managing returnable transportation items. Recently, Bressanelli, et al. [37] derived 16 challenges from previous research on CLSCs within the circular economy context.

**Table 1.** Comprehensive reviews that considered closed-loop supply chains (CLSCs) in their investigated domain.

| References | Domain and Contribution | Year | No. Paper |
|---|---|---|---|
| Agrawal, et al. [38] | Classification of the RL and CLSC literature regarding implementation disposition, return, and networks | 1986–2015 | 242 |
| Govindan, et al. [39] | Content analysis of CLSC and RL works to find the gaps | 2007–2013 | 382 |
| Schenkel, et al. [40] | Reviewing green, reverse, and closed-loop supply chain literature to synchronize existing knowledge on value creation | 1998–2014 | 144 |
| Bazan, et al. [41] | Reviews the mathematical models for RL and suggesting a potential modeling approach for green RL | 1967–2015 | 183 |
| Jena and Sarmah [42] | Reviewing CLSCs on remanufacturing with special emphasis on literature related to acquisition management of returned items | 2000–2014 | 100 |
| Diallo, et al. [43] | Classification of the CLSCs, with a focus on remanufactured or second-hand products | 1985–2016 | 104 |
| Rajeev, et al. [44] | Classification various factors considering the triple bottom line of sustainability issues (i.e., economic, environmental, and social) including CLSCs | 2000–2015 | 1068 |
| Gaur and Mani [45] | Focusing on CLSCs in emerging economies and suggesting the framework that includes seven driving forces for CLSC | 1992–2015 | 141 |
| Islam and Huda [46] | Categorizing the RL and CLSC of E-waste into designing and planning of reverse distribution, decision making and performance evaluation, conceptual framework, and qualitative studies | 1999–2017 | 157 |
| Larsen, et al. [47] | Identification of distinct opportunities and contingency factors in the area of financial performance for RL and CLSCs | 1995–2016 | 112 |
| Moreno-Camacho, et al. [48] | Identifying indicators used when sustainability is evaluated in real cases for forward, reverse, and closed-loop supply chain network design | 2015–2018 | 113 |
| Peng, et al. [49] | Analyzing the causes of uncertainties at different stages of CLSCs, and identifying appropriate methods for quantifying the impacts of the uncertainties on production processes | 2004–2018 | 302 |
| Present review | Categorizing, analyzing, and figuring the structure of the game-based CLSCs regarding selling and collection directions, type of the game, reprocessing, and planning horizon | 2004–2020 | 230 |

This research investigates two previously mentioned questions, leading to an in-depth analysis of game-based CLSC models addressed in more than 220 papers. We studied nearly 800 scenarios derived from the reviewed papers, including CLSC models based on GT. The papers were searched systematically, and the whole body of each paper was reviewed to determine the roles of participants and to study the collection and selling channels. Moreover, the investigated CLSC structures were surveyed from other perspectives such

as planning horizon and reprocessing activities, including remanufacturing, recycling, and refurbishing. As a novel work, the structures of the discussed models were graphically presented and analyzed, providing a unique, comprehensive, and compact view of the related literature along with many details about the developed models. Subsequently, directions for future research are suggested based on existing gaps.

### 3. Methodology

The methodology applied by Shekarian, et al. [50] and Shekarian [11] was used to gather the relevant research [51]. Specifically, it included material collection, descriptive statistics, category selection, and material evaluation.

### 3.1. Material Collection

The papers were derived using the Web of Science$^{TM}$ (WoS) search engine, previously known as the Web of Knowledge. According to www.clarivate.com, https://clarivate.com/webofsciencegroup/solutions/web-of-science/, "it is the world's most trusted publisher-independent global citation database and the most powerful research engine, delivering best-in-class publication and citation data for confident discovery, access and assessment." The terms "closed-loop supply chain", "closed-loop," and the abbreviation "CLSC" for the period of 2004–2020 were searched under the searching option "topic" provided by WoS. The term "closed-loop supply chain" is one of the high-frequency terms mainly used in green supply chains [52]. We considered the papers published in journals and written in the English language. The initial search, as described in the first part of Figure 1, led to almost 750 papers regarding the mentioned refined options. The keywords and abstracts of these papers were investigated to find relevant research at the filtering step. Unrelated papers that considered other sustainability areas were removed. We only concentrated on the research that developed and modeled a supply chain based on closing a loop. For example, papers that only studied reverse logistics and green activities and disregarded forward activities were excluded. This step led to selecting 460 papers, which are presented in the second part of Figure 1. At the analyzing step, each paper's content was reviewed to determine the papers that were designed and solved based on GT. In particular, we searched keywords such as "game," "Nash," "competition," and "Stackelberg" to extract the related papers. The CLSC models that were solved based on GT methods were investigated in detail. We found 180 papers that fit the scope of the present review, shown in the third part in Figure 1 called the "analyzing step".

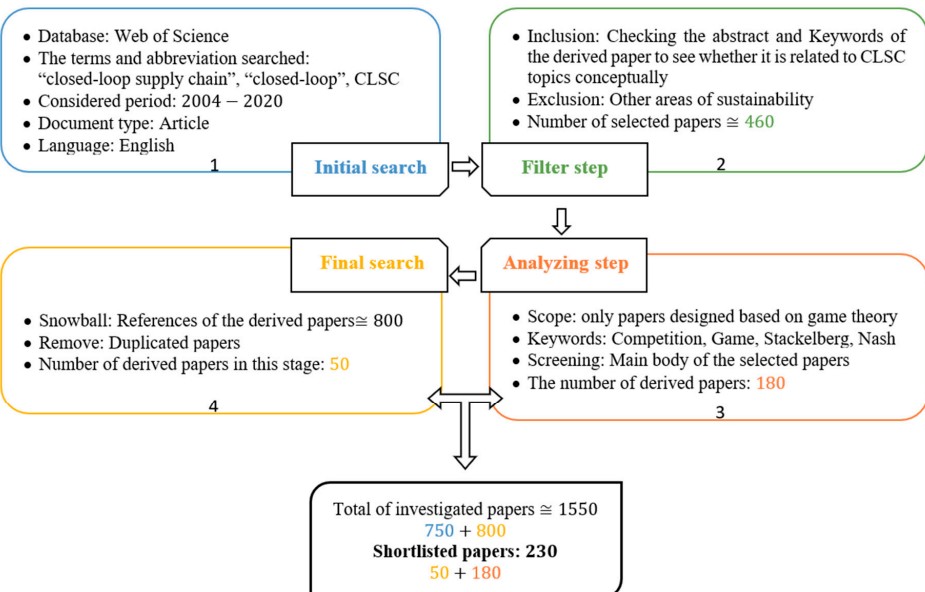

**Figure 1.** Illustrating the searching process.

Moreover, the references of 180 papers were snowballed to ensure that we did not miss any articles. By using this step and checking 800 more papers and after removing the duplicated articles, 50 more relevant papers were added, as explained in the fourth part of Figure 1. The last part of Figure 1 shows that these searching procedures finally resulted in 230 papers. The whole searching process is illustrated step-by-step in Figure 1.

### 3.2. Descriptive Statistics

As the research topic is interdisciplinary, we found a wide variety of journals with different scopes that published at least one related paper. In total, 64 journals were searched. Table 2 shows the distribution of journals that published at least two papers in this research scope annually, during the considered period. Figure 2 illustrates how many papers are published overall each year. The results confirm an increasing trend in the number of published works exploring CLSC based on GT, such that an average of 14 papers were published each year, which means almost one paper each month. We observed significant growth over the past few years, specifically since 2016.

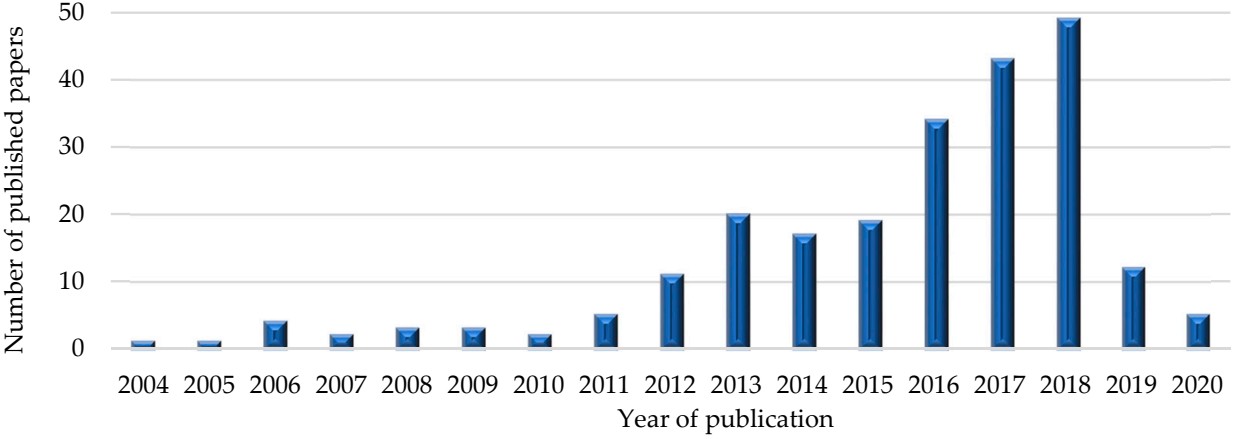

**Figure 2.** The number of published papers annually.

Figure 3 illustrates the number of published papers annually, considering the journals that published only one paper. Figure 3 can be interpreted as the measure of interdisciplinary of the field. The contribution of these journals is about 15 percent. These single-paper journals cover a broad discipline varying from mathematics, operations research, and artificial intelligence to economics, environmental, engineering, and management. Figure 4 shows the contribution of different countries based on the published papers. The first three countries are highlighted as China, the USA, and India, respectively.

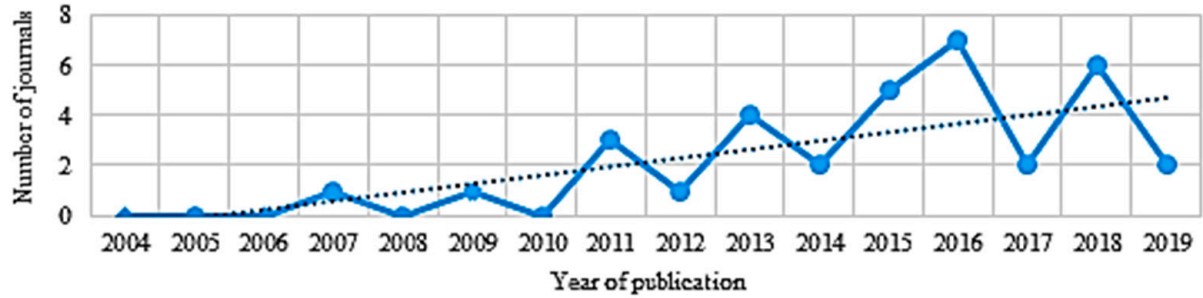

**Figure 3.** The number of published papers annually considering the journals that published only one paper.

**Table 2.** List of journals that published at least two papers during the considered period.

| Journals | 2004 | 2005 | 2006 | 2007 | 2008 | 2009 | 2010 | 2011 | 2012 | 2013 | 2014 | 2015 | 2016 | 2017 | 2018 | 2019 | 2020 | Total |
|---|---|---|---|---|---|---|---|---|---|---|---|---|---|---|---|---|---|---|
| Sustainability | | | | | | | | | | | 3 | 1 | | 5 | 16 | 2 | 1 | 28 |
| International Journal of Production Economics | | | | | 1 | | | | 1 | 2 | 2 | 4 | 4 | 7 | 1 | | 2 | 24 |
| Journal of Cleaner Production | | | | 1 | | | | | | | | | 3 | 6 | 8 | 1 | 1 | 20 |
| European Journal of Operational Research | | 1 | | | | | | | 1 | 2 | 2 | 1 | 3 | 3 | 4 | 1 | | 18 |
| International Journal of Production Research | | | | | | | 1 | | 1 | 2 | 1 | 2 | 2 | 4 | 1 | 1 | | 15 |
| Production and Operations Management | | | 2 | | | 2 | | | 2 | 2 | | | | 1 | 1 | | | 10 |
| Mathematical Problems in Engineering | | | | | | | | | | 1 | | 1 | 3 | 1 | 2 | | | 8 |
| Journal of Intelligent Manufacturing | | | | | | | | 1 | | | | 1 | | 2 | 2 | | | 6 |
| Transportation Research Part E | | | | | | | | | 2 | | | | 1 | 1 | | 1 | | 5 |
| Management Science | 1 | | 2 | 1 | | | | | 1 | | | | | | | | | 5 |
| Journal of Manufacturing Systems | | | | | | | | | | | 1 | 2 | | 1 | | | | 4 |
| Applied Mathematical Modelling | | | | | | | | | | 2 | | 1 | 1 | | | | | 4 |
| Computers & Industrial Engineering | | | | | | | | | | | | | 1 | 1 | | 2 | | 4 |
| Omega | | | | | | | | | | | 1 | | 2 | 1 | | | | 4 |
| Annals of Operations Research | | | | | | | | | | | 1 | | | 2 | | | | 3 |
| Asia-Pacific Journal of Operational Research | | | | | | | | | 2 | | | | | | 1 | | | 3 |
| European Journal of Industrial Engineering | | | | | | | | | | | | | 2 | 1 | | | | 3 |
| Abstract and Applied Analysis | | | | | | | | | | | 3 | | | | | | | 3 |
| Discrete Dynamics in Nature and Society | | | | | | | | | | 1 | | 1 | | | 1 | | | 3 |
| IEEE Transactions On Engineering Management | | | | | | | | | | 1 | | | | 1 | 1 | | | 3 |
| International Journal of Environmental Research and Public Health | | | | | | | | | | | | | | 2 | 1 | | | 3 |
| Industrial Engineering & Management Systems | | | | | | | | | | | 1 | 1 | 1 | | | | | 3 |
| International Journal of Advanced Manufacturing Technology | | | | | | | | | | | 1 | | | 1 | | | | 2 |
| Expert Systems with Applications | | | | | | | | 1 | | | | | | | 1 | 1 | | 3 |
| RAIRO Operations Research | | | | | | | | | | | | | 1 | | | 1 | | 2 |
| International Journal of Simulation: Systems, Science & Technology | | | | | | | | | | | | | 2 | | | | | 2 |
| Journal of Systems Science and Systems Engineering | | | | | | | | | | | | | | 1 | 1 | | | 2 |
| Journal of Renewable and Sustainable Energy | | | | | | | | | | | | | | 1 | 1 | | | 2 |
| Industrial Management & Data Systems | | | | | | | | | | | | | | 1 | 1 | | | 2 |
| Mathematical and Computer Modelling | | | | | 1 | | 1 | | | | | | | | | | | 2 |

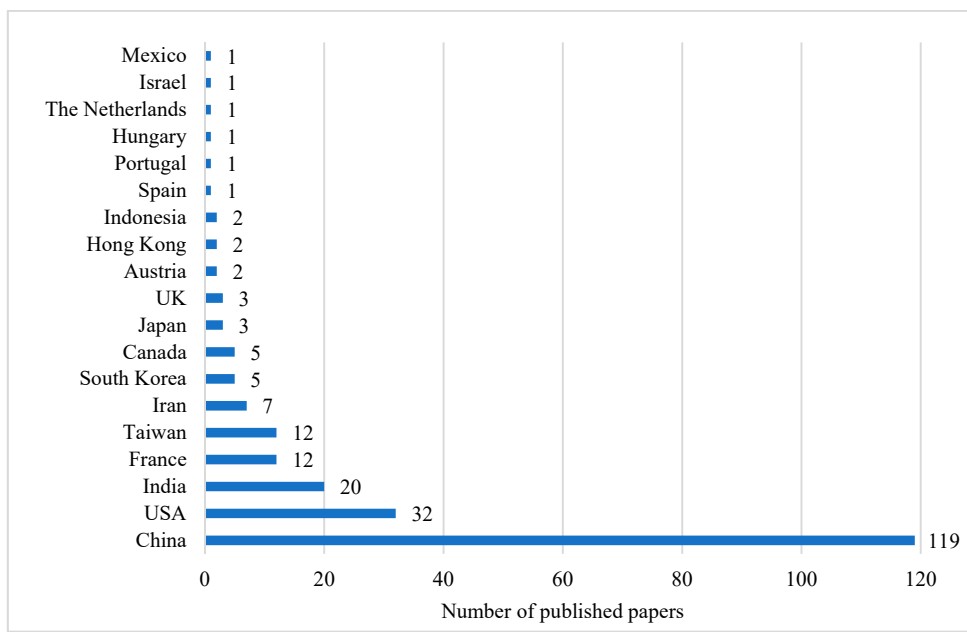

**Figure 4.** The contribution of different countries to the investigated field.

### 3.3. Category Selection

To reply to the research questions and expand the topic through the literature, we classified the selected papers into various categories by applying a systematic analysis. The major categories were "selling and collection channels," "type of game," "reprocessor party," and "planning horizon." Each major category is composed of a few subcategories. It is a top-down classification method. For example, under the category "channel," papers were studied considering the number of involved parties in collection and selling activities. In the next layers, they are divided into "single-channel" and "multiple-channel" subcategories. Figure 5 shows a detailed overall view of the complete research.

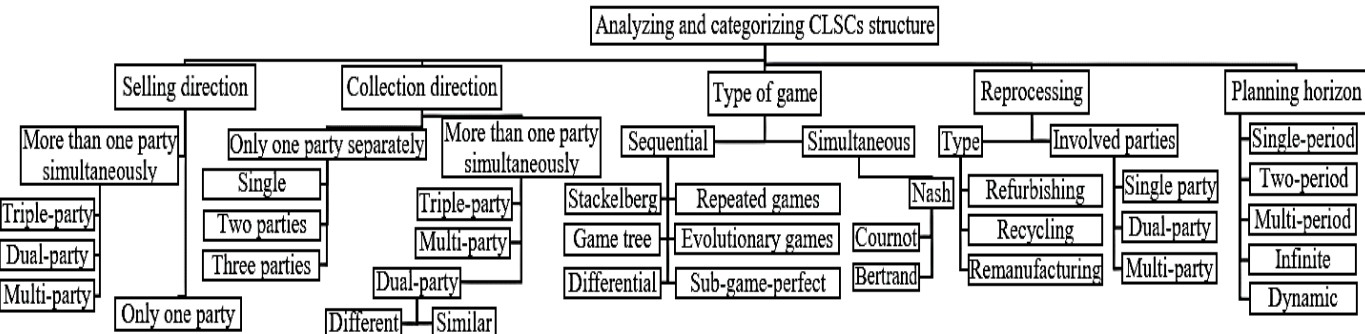

**Figure 5.** Categorization of the discussed CLSC structures.

### 3.4. Evaluation Stage

Papers derived in the previous stages were crossed-checked with the results in Scopus, the most comprehensive academic database to ensure the reliability of the whole process. Moreover, a snowball was implemented on the selected papers' references to ensure that all the related papers were considered.

## 4. Results

The overview of structure and categorization is presented in Figure 5. In addition, Table S1 in the supplementary materials shows details results of our literature study. The references are sorted based on the year of publication and the family name of the first author. An empty field in Table S1 indicates that this field does not apply. In the following, we explain the results based on the content of the reviewed papers.

### 4.1. Closed-Loop Supply Chains (CLSC) Networks

In this section, we study the design of a CLSC regarding the forward and reverse directions. On the basis of the structure of the collection in the investigated CLSCs, the collection method is categorized into two general classes, as shown in the second and third columns in Table S1. The first category uses a single-collection method in which only one-party manages to collect the cores (used items) and, in the second category, at least two parties are involved in the collection [53]. The multiple-collection method also covers the prevalent method of so-called dual-channel collection in which two parties simultaneously collect the used items [54,55]. Figure 6a,b show a typical single and dual collection system. A similar categorization could be found for the selling process, as illustrated in Figure 6c,d.

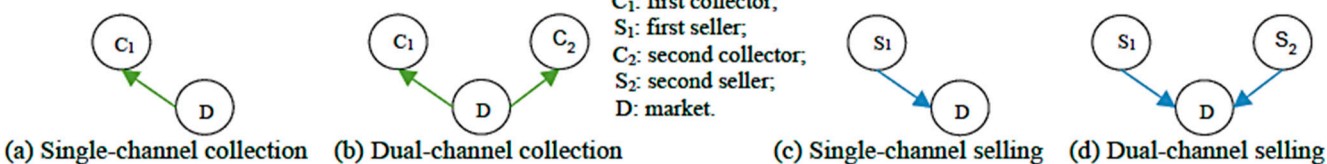

**Figure 6.** Single- and dual-channel selling and collecting.

Moreover, the papers that use the single-collection method can be classified into three subcategories. The first subcategory considers only one party as the collector [56–58]. However, the second subcategory includes those papers comparing collection by two parties separately, such as Li, et al. [59] who assumed either the manufacturer or the retailer was the collector. The third subcategory studied the collection by more than two parties separately [60,61]. The single and multiple channels are indicated by "&" and "-" in the related columns in Table S1. It is evident among the different types of collections that the single collection method is the most frequently used method.

The earliest work that compares the performance of a CLSC using GT regarding the different single-channel collection is that of Savaskan, Bhattacharya and Van Wassenhove [12] who reported, ceteris paribus, that the player who is closer to the customer is the most effective undertaker of product collection activity. After that, many scenarios are suggested for collection. Savaskan and Van Wassenhove [62] later verified that such a result is still valid even when retailers compete on prices. Researchers extended the seminal work of Savaskan, Bhattacharya and Van Wassenhove [12], by discussing the impact of collection cost structure on the manufacturer's optimal reverse channel choice [63,64]. Furthermore, other works discussed single-channel collection under the separate channels of a retailer, manufacturer, and third party (M&R&T) considering fuzzy environment [65], two-period setting [5], local advertising [66], the reference price effect [67], and product quality level [68].

The dual-channel collection method has received priority in the category of multiple-collection systems, due to practical motivations, such as Xerox that collects cores through retailers and provides prepaid mailboxes for customers to return the cores directly. Some papers compared the effects of different dual channels, such as Zhao, et al. [69] who studied a manufacturer's collection by either a retailer or third party. Hong, et al. [70] examined different hybrid channels and found, ceteris paribus, that the manufacturer and the retailer dual collection channel (M-R) is the most effective reverse channel structure for the manufacturer and is superior to the single-channel collection approach. These results

were confirmed by Liu, et al. [71] regardless of the competition intensity. Motivated by the collection system ReCellular Inc., the largest cell phone remanufacturer in the USA, Hong, Zhang, Zhong and Liu [54], as a novel idea, investigated retailers' advertising decisions regarding hybrid dual-channel collection problems.

Some works only focused on investigating a specific dual-collection system by combining two different parties or similar parties. Regarding the latter one, dual collection by two retailers is the most studied case [72,73]. Different parties that collect cores are also considered simultaneously, such as collecting electrical and electronic equipment by a retailer and a third party (R-T) [74] or collecting through a physical sharing strategy in which both the manufacturer and a third party (M-T) are responsible for the collection process [75].

A combination of more than two parties in the collection process makes the problem even more complicated. Bernard [76] studied a triple-collection process by a duopoly of identical original manufacturers and an independent remanufacturer (IR) in the aftermarket by recovering and remanufacturing used products. Ma and Chen [77] considered three oligarchy retailers that collect used products and recycle waste products. Chu, Zhong and Li [53] extended the previous models [12,62] by developing a Stackelberg game model and showed that the joint third-party collection mode serving multiple manufacturers can outperform individual retailer- and manufacturer-managed modes. A practical case of this case is Taolv365 (www.taolv365.com) that offers an Internet-based takeback service in the Chinese market to mobile phone manufacturers, including Samsung, Lenovo, and TCL.

In fifty-eight models, the single-channel selling was extended to dual channels [78–80]. Different dual-selling channels could be divided into several classes when a manufacturer is paired with a remanufacturer (M-Re) [81], a retailer (M-R) [82], a third party (M-T) [83], or another manufacturer ($M_1$-$M_2$) [84], and a retailer is paired with a third party (R-T) [85] or another retailer ($R_1$-$R_2$) [86]. According to the results, the most attention is given to the dual-selling channel managed by a manufacturer and a remanufacturer. Moreover, in Table S1, there are online selling channels by a manufacturer or retailer paired with an offline retailing channel of a retailer shown as (eM-R) [87] and (eR-R) [88], respectively. In contrast to the dual collection cases, online selling received more attention such that 16 models studied cases of dual selling where the manufacturer sells its product online simultaneously with the retailer. It should be noted that 34 models designed both dual selling and dual collection at the same time.

### 4.2. Game-Based CLSCs

The investigated games in formulated CLSCs can be categorized into two general categories. The first category includes the papers dealing with models of sequential games, and the second category is comprised of papers dealing with models addressing simultaneous games. The results are presented in the sixth and eighth columns in Table S1. In order to provide detail, the involved parties are mentioned in each class. It should be noted that, in the sequential games, we only mentioned the first player who moves as the principle one. Moreover, the investigated models are identified based on the specific games presented in the eighth column in Table S1. Various games are addressed in the reviewed papers, including Stackelberg, Nash, Bertrand, Cournot, differential, and evolutionary games.

Most of the models in the first category are designed based on the context of Stackelberg duopoly/competition, a model of imperfect competition based on a non-cooperative game. In this game, one firm, which is probably better known or has greater brand equity, is the first to decide to be the so-called leader. The follower(s) observe(s) this and move(s) later. Nevertheless, other types of games are identified in the framework of the Stackelberg model. We can refer to [6,89,90], who worked to formulate CLSCs based on differential games. These games are solved by the Hamilton–Jacobi–Bellman equation method [91,92]. The evolutionary game, a population model with a game embedded in it, is another type that is used to study companies' behavior and find evolutionary stable strategies of players in the long term [93,94]. Some researchers used dynamic GT that considers the action be-

tween the participants in order (i.e., the latter player observes the former player's behavior choice, and then makes an appropriate choice according to this) [84,95].

In addition to the usual duopoly Stackelberg, other types of sequential games are used. As a practical case, sequential games are studied in the form of a game tree, for example, analyzing producers' and consumers' behavior regarding choices between refillable/return and disposable bottles [13] and computer waste [96,97]. The integrated multiple manufacturing/remanufacturing cycles CLSC inventory system is formulated, extended, and solved sequentially with multiple players when the manufacturer is the first reactor [98–101]. Moreover, repeated games, also known as supergames, as another form of sequential games have been studied. Supergames are defined as an extensive form of a game consisting of a number of repetitions that parties play out over and over for a period of time. They are used to study the pricing policy of a manufacturer and remanufacturer under the case of competition [102], analyzing complex dynamic phenomena such as bifurcation, chaos, and continuous power spectrum in a CLSC [103,104].

The performances of CLSCs can vary under different channel leadership as it influences essential parameters of the system, such as the equilibrium of retail and wholesale prices, return rate, and the effort of "take-back." Which leadership model is the most effective model is an important question that is discussed in different loops, due to the fact that members of CLSCs can benefit from being the dominant player [105], for example, obtaining more profit and affecting the products and decisions of other players. Therefore, identifying the leader and dividing the power in this structure is essential. Channel power is defined as one party's ability to prompt other parties to do what they would not have otherwise done [14]. In the form of a game, when a specific party (i.e., retailer) has significantly greater power, it is the party (retailer)-Stackelberg game.

Most of the papers considered the leadership effect of only one party in the developed CLSCs (e.g., manufacturer [12,106,107], supplier [108–112], retailer [113–115], remanufacturer [116,117]). The studies prioritized the manufacturer's role as the leader, such as in real situations (e.g., national manufacturing brands), they are often bigger as compared with other players and probably have stronger risk tolerances. Different CLSCs are studied under manufacturer-leader to investigate different topics such as the role of the players as marketing investor [118], recycling fund system [119], and the effect of risk [120]. However, some models compare the effect of two or three leaders (see, for instance, [14,58,121–125] and [15,66,105,126,127] that consider triple and dual leaders, respectively). However, if the power difference between the parties is equal or minor, it can be studied through a Nash game.

The CLSC in which parties compete simultaneously and non-cooperatively in a formulated non-cooperative game has been investigated via Nash cases. In these cases, we can classify the models according to the competitor parties. Strategic planning and recovering and remanufacturing decisions between a manufacturer and remanufacturer are discussed under different situations such as period-dependent settings, subsidy and take-back laws, and technology licensing [55,128–137]. There are efforts to derive Nash equilibrium when there is competition between two independent manufacturers considering trade-in strategy [138], market segmentation [139], local and non-local manufacturing [140], product life cycle [141], recycling choice [142], to name a few. Competition between a retailer and manufacturer through a Nash framework is addressed by the retailer reselling refurbished cell phones in the second-hand markets [143], contracting reverse revenue sharing [89], and influencing government intervention [144]. Moreover, other Nash-based designed games between other parties such as a supplier-manufacturer (for example, see [145], [146] and [147] for joint economic lot size, environmental impact, and supply risk problems in CLSCs, respectively) and manufacturer-third-party (for example, see [148] for cannibalization problem and [149] for optimal relicensing fee issue) are solved in the literature.

Some studies have attempted to find the sub-game-perfect Nash equilibrium in the Stackelberg-based games through a backward induction [150–152]. This concept is used to determine the appropriate strategy (for instance, outsourcing or authorization [5,153],

remanufacturing [76], relicensing [149], product design [154], bricks and clicks [87]) among the different parties. Sabbaghi, et al. [155] studied the behavior of consumers and original equipment manufacturers (OEMs) based on an on-time return of the WEEE, using a series of equilibrium strategies. Yoo, Kim and Park [111] derived pricing and return policies under various supply contracts between a supplier and retailer. Yoo and Kim [156] derived the relevant equilibrium between new and refurbished items in a three-echelon CLSC. Subgame perfect equilibrium is combined with alternative bargaining offer to overcome the deficiency of transfer of pricing policy [68].

Regarding the second category, there are two well-known games which are Cournot and Bertrand duopolies. Cournot duopoly is an imperfect competition in which firms with identical cost functions compete with homogeneous products in a static setting, which happens when output and capacity are difficult to adjust. It is applied for different situations such as choosing sales quantities for multiple firms acting under efficient take-back legislation [157], studying the value of information-intensive product recovery systems under a quantity-competition market [158], competing the duopolies retailers to optimize sales prices and recycling rate [159], and obtaining sales quantities in a competition between a manufacturer and remanufacturer considering a re/manufacturing system [8,83,160]. The Bertrand duopoly, alternatively, occurs when a strategic choice is based on prices, rather than quantities, especially in the short term. This type of game is studied when two manufacturers are selling new products through a common retailer [161], when a retailer and third party are making simultaneous decisions under an advertising condition [54,66] and dual recycling channel [71], and when an OEM and IR are determining the price of a new and a remanufactured/refurbished product [162,163], to name a few.

For the interested readers, all the scenarios derived from the papers [3–8,12–15,53–105,107–111,113–269] are briefly categorized in the Supplementary Material (Table S1) provided online.

### 4.3. Reprocessing

In our investigation, three main reprocessing concepts, including remanufacturing, recycling, and refurbishing (3R), are taken into consideration in the reviewed models. Among them, remanufacturing is the one with the most priority. The remanufacturing process provides a unique opportunity for companies, on the one hand, to improve their profits, and on the other hand, to serve social responsibility [196]. For example, Ford collected more than 332,000 pounds of toner cartridges for remanufacturing and saved USD 1.2 million from 1991 to 1997 [131]. The column called "RP" in Table S1 indicates which party is responsible for these activities (3R). If at least two parties are involved simultaneously, they are represented with a "-" (e.g., see M-Re in [134]), and if in different models in a paper different parties are considered, they are separated with a "&" (e.g., see T&Sup in [151]). Some models also focused on the trade-in process, i.e., replacing a fraction of the collected cores by new ones [3,95,227,241].

The results prove that efforts to develop the CLSCs with a single-reprocessor (SP) party are most often taken, and among them, the manufacturer took priority over other parties as a remanufacturer [211,270], recycler [159], and refurbisher [238]. OEMs are widely encouraged by business leaders, environmentalists, and social planners for engaging in remanufacturing. For instance, Xerox, Hewlett Packard Corporation, and Canon have been a leader in reusing their high-value and EoU copiers, computers, and cartridges, respectively, to manufacture new items with the same quality standards. However, other parties such as suppliers [108,172,271], remanufacturers [132,205], and retailers [57,254] are also considered to be the reprocessors individually. CLSCs are compared under different SPs such as Hong and Yeh [170] and Xiong, Zhao and Zhou [212] who developed models with either a manufacturer or supplier (M&S) to study a different type of collection in the electronics industry or to investigate environmental performance in the remanufacturing industry, respectively.

Regarding the channel design in processing, another category of papers considered two parties that are able to do reprocessing simultaneously. Two cases, i.e., manufacturer-remanufacturer [129] and two manufacturers $M_1$–$M_2$ [140], are the most common ones in the literature. In fact, reduced production costs attract IRs. Considering the first case, CLSCs are designed with two manufacturers that remanufacture the collected items without [161] and with [244] incentive mechanisms provided by the government [228]. The competition between two manufacturers was considered in order to study the effect of recycling strategy inspired by the metal cutting tools industry by Raz and Souza [142] and the impact that the brand of a remanufactured product has on total profit by Jena, Sarmah and Sarin [261]. In 2005, IRs represented 54% of the aftermarket for automotive parts in Europe and 66% worldwide [76]. The competition between a manufacturer and remanufacturer occurs due to consumers' desire for environmentally friendly and economical products [162], the status of take-back regulation for recycling [8], and to prevent the effect of cannibalization [128].

Many OEMs allow a third party to perform 3R operations of branded or patented products through outsourcing or authorization [153]. This constitutes another category in dual-party reprocessing studies. For example, Kodak Inc. signed a licensing agreement with remanufacturers to provide remanufactured inkjet cartridges that are compatible with Kodak's printers [160]. Liu, Lei, Huang and Leong [163], and Oraiopoulos, Ferguson and Toktay [149] examined the competition models between an OEM and a third party to study the effect of refurbishing strategy interfacing the secondary market of the electrical and electronic products and the information technology industry. Moreover, Huang and Wang [218] developed a CLSC that manufacturer remanufactures a fraction of used products and licenses the third party to remanufacture the rest. Other models have been formulated when a manufacturer and supplier [201] or a manufacturer and retailer [188] are engaged in remanufacturing.

*4.4. Planning Horizon*

One of the essential assumptions in most CLSC models based on GT is that players of the loop make their decisions in a single period setting with the previous existence of the product in the market [12,66,137,232]. As presented in the ninth column of Table S1, we searched this assumption through the models. Single-period models represent the maturity stage of a product's life cycle where prices, as well as return and remanufacturing rates are steady [8]. As a slice of an infinite horizon model, they are applicable to items such as electronic products, where the manufacturers are less interested in keeping inventory because of high obsolescence rate and rapid decline in price over time [221]. Products sold in previous periods can be returned to the manufacturer for reuse unconstrained, and the manufacturer can maximize her profit by choosing an arbitrary production quantity of the remanufactured product. The majority of the models are consistent with this assumption as it facilitates analytical tractability without the distraction of initial and terminal time-period effects [83].

Another group of papers (almost 15%) developed CLSC in a two-period setting. The competition in the second period happens because the product sold as a new one in the first period can be resold as a remanufactured or refurbished product by the previous party [4,78,79,252] or a new party [55,128,133,134,149,181]. Concerning this matter, De Giovanni and Zaccour [5] and Su, Li, Tsai, Lu, Liu and Chen [250] extended the static model by Savaskan, Bhattacharya and Van Wassenhove [12] to a two-period framework game by considering the option of outsourcing for a manufacturer. Two-period models are also formulated to include a trade-in option. In the first period, consumers buy the new products, while in the second period, they can choose to continue using the used product or replace it through trade-ins [3,138,141,247]. The motivations for proposing the second period include introducing an upgraded version of a product [251], consideration of a warranty period [238], selling separation of new and remanufactured items as online and offline channels, respectively [85], selling remanufactured products at another loca-

tion [202], cores supply limitation in the second period [139], the effect of quality in the first and second periods [259], and studying cannibalization issues [148].

Managing a CLSC with one type of reprocessing (i.e., remanufacturing, recycling, and refurbishing) is typically a multiple-period problem as a new product should be used for a determined period [165]. However, there has been little effort to deal with this issue applying GT in CLSCs. A multi-period is applied regarding the product, for example, consumer behavior dealing with the defective product [76] or the structure of the CLSC, for example, maximizing returns in discrete time periods [103,185]. A real case of the latter example was investigated by Gu, Ieromonachou, Zhou and Tseng [135] and Gu, Ieromonachou, Zhou and Tseng [136] as a three-period CLSC model for making, sorting, and recycling of electric vehicle batteries in the first, second, and third periods, respectively. Considering the former case, Gan, Pujawan and Widodo [190], and Gan, Pujawan, Suparno and Widodo [217] studied a four-period model for a short lifecycle product with an obsolescence effect after a certain period.

According to our results, dynamic CLSC models, which are more applicable and fitted to reality (see, for example, De Giovanni [72] who explained battery CLSC dynamically) have been rarely used in the literature [203]. A dynamic process can reflect the applicable cases as the parameters of a CLSC, such as demand and quality [92] or inventory level of players [6,182], are usually a function of time. In addition, it provides an opportunity to examine the effects of planning horizons on the equilibrium results [160] or important elements such as warranty period [266]. There are models that are developed in an infinite planning horizon [94,98].

*4.5. CLSC Structures*

This section uniquely and graphically reviews and illustrates the discussed CLSC models. Regarding the differences in structure, applied methods and concepts, and format of the game, 196 different CLSC structures are depicted in Figure 7. In order to make it as concise as possible, the structures are drawn for the most general cases that include the designed subsystems. For example, in Structure 109, derived from [107], we only figured the decentralized case as the centralized one is clear. The composing elements of these figures are defined in Table 3. The last column called "Figure" in Table S1 determines the relevant figure(s) to each reference.

**Table 3.** Definition of notations and arrows in the CLSC structures presented in Figure 7.

| Arrows | Definition | Notation | Definition | Notation | Definition |
|---|---|---|---|---|---|
| → | Forward channel | D | Market | DP | Primary consumer |
| → | Reverse channel | SM | Second-hand market | DR | Replacement consumer |
| ⇒ | New product channel | M | Manufacturer | AD | Advertising agency |
| ⇒ | Remanufactured product channel | R | Retailer | (party)* | It shows the party is able to remanufacture the cores |
| → | Disposal to landfill | Re | Remanufacturer | (party)# | It shows the party is able to recycle the cores |
| → | Trade-in channel | T | Third-party collector | (party)f | It shows the party is able to refurbish the cores |
| - - → | Transferring the information | S | Supplier | $(party)^T$ | It shows the party is able to replace the cores |
| → | Government's effect | g | Government | r | Licensee |
| → | Outsource advertising services | L | Landfill | e | Online channel |

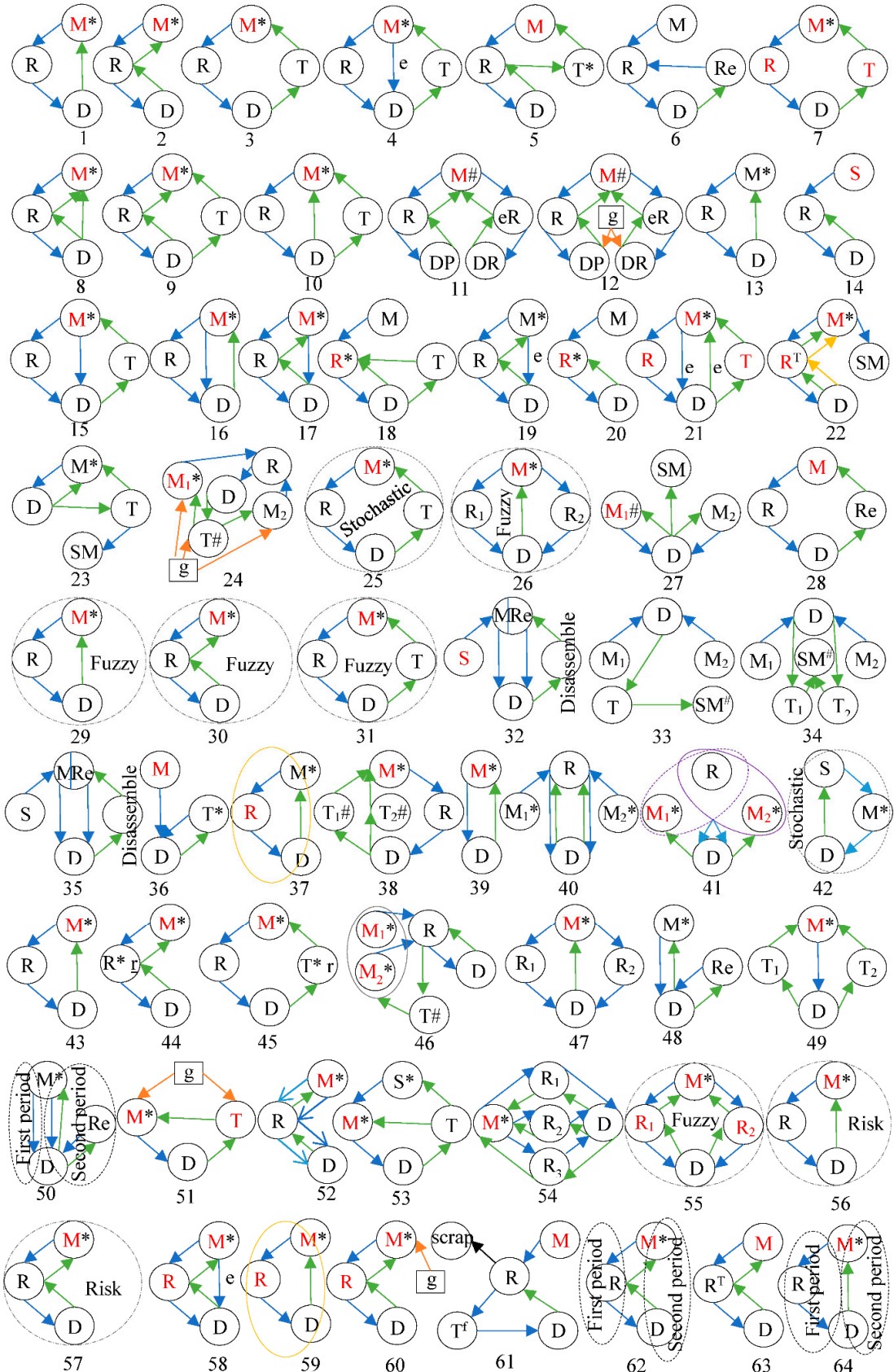

**Figure 7.** *Cont.*

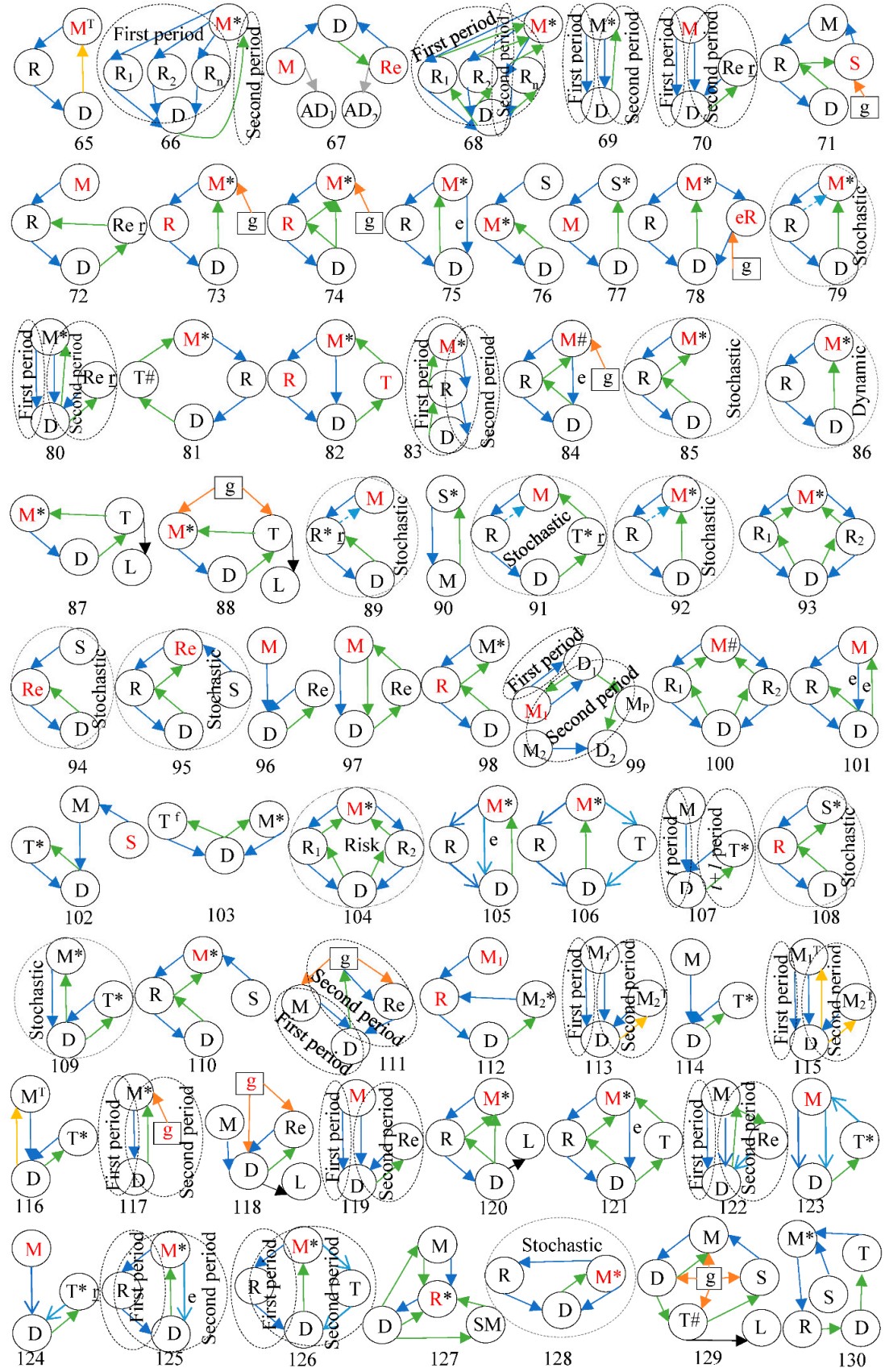

**Figure 7.** *Cont.*

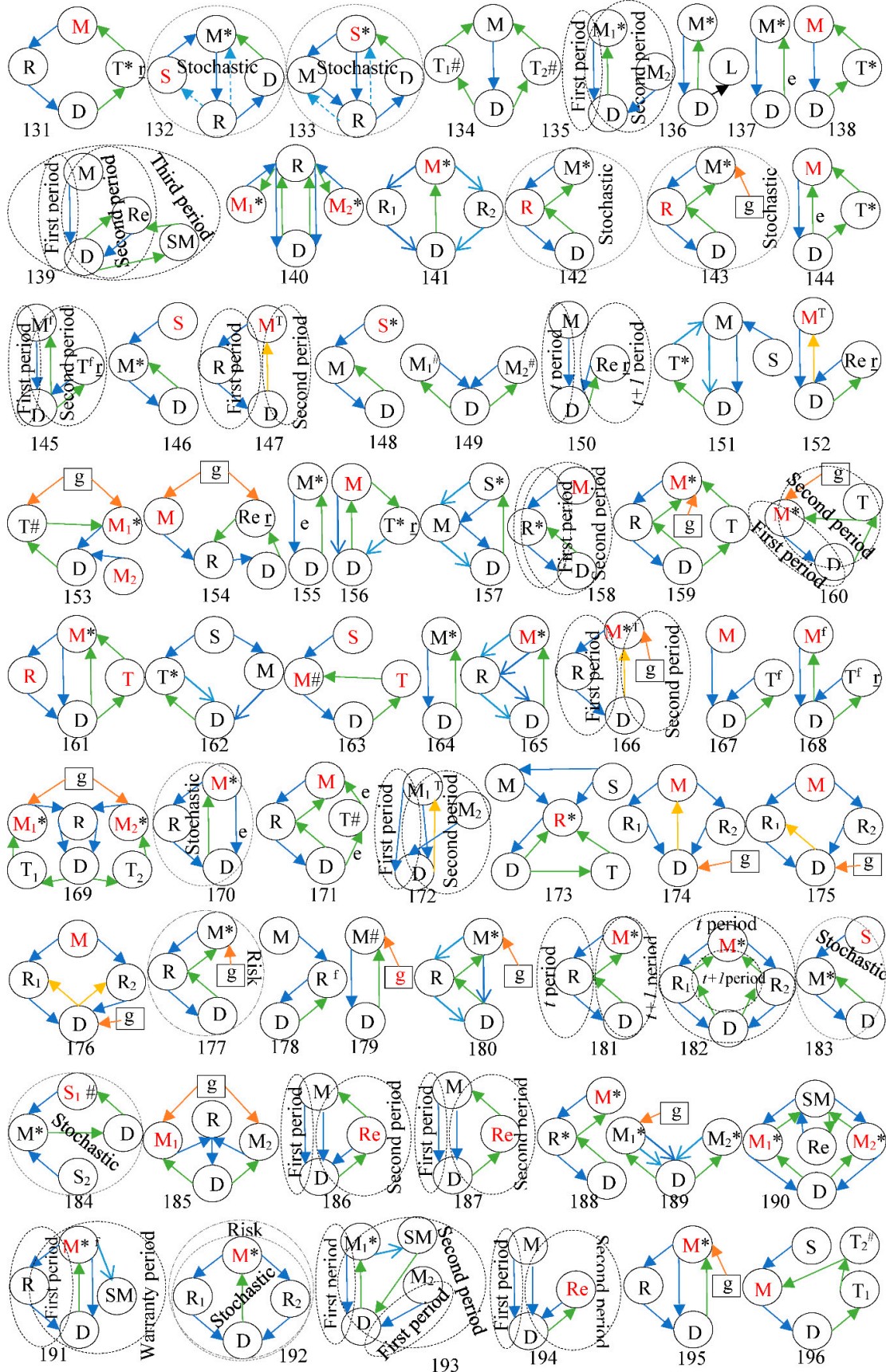

**Figure 7.** Graphical review of the CLSC structures.

Three specific reprocessing options (i.e., remanufacturing, recycling, and refurbishing) are determined with the symbols "*", "#", and "f", (for instance, see Structures 15, 16, 17 [208]; 196, 201 [260]; and 167, 168 [163]), respectively. As shown, these processes are undertaken by different parties (e.g., M* means the manufacturer can do remanufacturing). Among these parties, a third party mainly plays the role of a collector. However, a third party can also act as a recycler [260], refurbisher [156], or remanufacturer [222] (e.g., see Structures 196, 61, and 102, respectively). The parties are connected with the market (or consumers) presented as D. Some models considered the second-hand market (SD) (see Structures 22, 34, 127, 139, 190, and 191, to name a few). For example, based on Structure 193 related to Mitra [139], the primary customers of the first manufacturer ($M_1$) may become environmentally conscious and substitute the $M_1$'s new products with its remanufactured products. In contrast, secondary customers, who are price sensitive, would never replace remanufactured products with more expensive new products. Some models also separate primary and replacement consumers shown as DP and DR, respectively. Replacement consumers referred to consumers interested in replacing their used items with new ones (see Structures 11 and 12).

The players are connected via colored arrows, as shown and defined in the first and second columns, respectively, in Table 3. In addition to the usual forward and reverse channels, more details are illustrated using different arrows. In some CLSCs, new and remanufactured products are sold to the customers either through different channels [151,153,249,258] (for example, see Structures 52, 123, 151, and 165 in Figure 7) or different parties [83,152,246,264] (for example, see Structures 105, 141, 156, and 162 in Figure 7). Few models considered the landfill effect shown with the black arrow [13,207,230] (see Structures 118, 129, and 136 in Figure 7). There are models concerning the effect of government (g) to design a reward-penalty mechanism for remanufacturing [60], government subsidy [94,177,241], carbon tax [247], incentives for returned products [228,244], and cap-and-trade regulation [82]. This effect is displayed with the red color (for instance, see Structures 60, 74, 88, 154, 166, 169, 175, 185, and 195 in Figure 7).

With the growing use of the Internet, players use online channels in CLSC [219,265]. These channels are determined with "e" in the reviewed models (for example, see Structures 85 and 170, in Figure 7). Furthermore, other characteristics such as transferring the information between the parties are presented with a dashed-blue arrow, as investigated by [61,220,232] (see Structures 79, 89, 91, 92, 132, 133 in Figure 7). These studies analyzed sharing the information among the players. For example, whether the retailer shares the private information of the market with a manufacturer is vital in a loop. Another highlighted feature is the trade-in process, defined as an exchange offer channel that the party applies to replace a fraction of the collected used products with new products. This replacing mechanism of a used product is shown with an orange color, and the relevant party is identified with "$^T$" (see Structures 22 [227], 65 [3], 116 [200], and 176 [241]). In some cases, a party is licensed by the manufacturer to do one of the 3Rs [7,78,134,149,218]. These parties are indicated with the symbol "r" in depicted CLSCs (e.g., Structures 44, 45, 70, 72, 80, and 145, in Figure 7).

We provide insights into how game-theoretic modeling frameworks are used through the CLSC. In these structures, entities that are considered to be leaders are shown in red. To prevent repeating, we showed all considered leaders in one structure. For example, the loop presented in Structure 7 shows that it is examined under three parties' leadership (i.e., manufacturer-led, retailer-led, and third party-led). Moreover, the structures with no party and shown in red are related to the simultaneous game. It should be noted that Structures 37 and 59 are related to different models. Purple ovals in Structures 41 and 46 show that integrating the internal parties in the developed models is also studied through the game.

Most of the literature discussed CLSCs based on deterministic and static models. However, there are models that have been developed in stochastic [91,187,192,245], fuzzy [168,240], and risk [114,257] environments. These models are distinguished in the drawn figures (see Structures 128, 142, 170, 184; Structures 26, 29, 30, 31; and Structures 56, 57, in Figure 7, respectively).

Moreover, in order to make a connection with the previous section regarding the planning horizon, we have shown the activities that are time dependent if a model is not static. As a sample, Structurer 122 demonstrates that the manufacturer sells the products to the market in the first period. Then, in the second period, the channels of new and remanufactured products are separated while a remanufacturer is involved and fed by the collector of the cores (i.e., the manufacturer).

In addition to the mentioned specifications, other factors are considered, such as the existence of outsourcing advertising efforts to agents (see the gray arrows), disassembly section with two split manufacturing-remanufacturing sections, scrap section, and warranty period, as displayed in Figure 7, Structures 67, 32, 61, and 191, respectively.

## 5. Discussion

### 5.1. Academic Implications

The majority of the models consider manufacturers as the parties who produce sustainable items and play the role of collectors. Although retailers and third parties can participate in the collection process, we understand that the manufacturer is the primary collector in the investigated models, which is consistent with some researchers' findings who suggested that outsourcing the product collection is an applicable option only when an outsource performs operationally and environmentally better than the manufacturer [5]. Comparing the manufacturer and retailers' performances on collection took precedence over the models that focus on different single channels [3,4,15,94,114,156,171,258,267]. In addition to the common entities, suppliers can play the role of collectors when there is a risk of insufficient sources [147,192] or preference of supplier remanufacturing because of the environmental impact [151,212]. In the case of the triple single-collection channel, most studies were devoted to comparisons among manufacturers, retailers, and third parties [63,65,194,250]. Regarding hybrid collection approaches, the literature showed that under the same conditions, retailer-manufacturer is superior to the single-channel collection ones [69,70].

Within GT's framework, two cases of particular interest for analyzing CLSCs are Nash and Stackelberg equilibriums. Nash equilibrium happens in decisional states from which no player is interested in departing, and the negotiation process reduces to a one-shot exchange of information. However, in many real situations, the equilibrium is not unique, and the first player, as the Stackelberg leader, imposes the game's outcome. The particular equilibrium reached in such an asymmetric game is called Stackelberg equilibrium [123]. We reviewed the literature regarding these two cases highlighting the player who is the leader in the adjusted cooperative game and the other involved parties. We found that almost three-quarters of the investigated CLSCs studied the models under the leadership of the manufacturers. Some works found a disadvantage by considering the manufacturer as a leader, such as the lowest return rate [90] or a higher retail price and a lower collection effort [121]. However, others showed advantages from the perspective of the remanufacturing process and consumers' welfare [58]. According to the results in Table S1, retailer-led models ranked second in almost 13% of the studied CLSCs. Furthermore, we found three models that considered the social planner/government as the loop leader. The manufacturer-led model was compared with the retailer-led model more than the other cases. Almost all the surveyed models in this area compare a manufacturer's performance as a leader with the leadership of the other parties.

According to our investigation, most of the studied models are steady-state models. However, the literature shows that non-static models are more fitted to treat the inherent attributes of CLSC, such as handling the differentiation of new and remanufactured products in a two-period model [57,85,130] or considering the life-cycle of a product [190]. Although the results of a few papers are examined by extending from a single-period to a two-period setting [153,166] or multi-period setting [129,154], extra efforts are required in this area.

Finally, the CLSC structures provide a better understanding of trade-offs using GT, CLSC, and other concepts, and the readers see and compare all the CLSC's structures

through a comprehensive frame. It shows that remanufacturing got priority among the other forms of sustainability. The results reveal that 13 percent of the papers formulated CLSC models based on GT in stochastic, fuzzy, or risk models. Cases in which two uncertain concepts were studied are rare [120] (see Structure 192). Therefore, there is still potential to formulate uncertain loops in the future.

Considering social planner's and government's interaction with players, we found that only 15 percent of the papers remarked on this role. A government's intervention towards the circular systems can affect only one player [74,94,171] or at least two parties [60,84,97,177,195,204]. We see that the manufacturer and the retailer, along with other parties, usually are in a vertically integrated game-based system. However, there are models that the players are competing or cooperating in a horizontal level [4,62,72,73,76,77,119,206,241,244,257] (for example, Structures 46, 54, 68, 93, 104, 169, 176, 182, and 190).

*5.2. Practical Implications*

Because of the destructive role of a third party for an OEM, retailers are highlighted as collectors, as there are companies such as Sony and Dell, who have established special collection plans such as GreenFill and Staples, respectively, providing facilities for their retailers to collect the cores. The literature proves that a third party collection model is usually ineffective in practice in a general CLSC [170]. Although remanufacturers are expected to be a major group of collectors, the interesting point is that they are not among the main parties responsible for collecting. Specifically, collecting used items is problematic for remanufacturers active in the area of WEEE, since collection basically depends on consumers' decisions on when and how they want to discard their cores [155]. There are examples for such cases in practice as Samsung collects its electronic products by offering the consumers a free mail-back option and a permanent drop-off option at over 200 locations. The competition could amplify the performance of collecting past-sold products in the case of dual designs [71]. As an example of a retailer and a third party collecting, ReCellular Inc., a recycler and reseller of cell phones in the USA, collects used phones from cellular airtime using a hybrid collection. With a combination of two retailers or manufacturer-third party, the dual-collection method is of interest to researchers, among others. In fact, the manufacturer's best action for collecting used products is to ensure the retailer is engaged in collecting used products regardless of adopting single collecting channels or dual collecting channels [69], and they could gain more core returns under the dual-designed channel [57].

Traditionally, manufacturers enjoy enough power acquire the lion's share of the supply chain profit and become a channel leader. In this case, they anticipate the retailer's response by making early decisions and offer supply contracts to the retailer [121]. The real world is the witness of such cases as seen in the recycling programs of giant OEMs such as General Motors, Toyota, Apple, HP, Lenovo, and Xerox. They play a more dominant role than their suppliers and their downstream members (see [272]). Although many previous studies assume that the manufacturer is the channel leader, there are real cases where huge retailers such as Carrefour, Wal-Mart, ToysRus, Gome, A&P, Tesco, and Hudson's Bay have excellent market power. With the expansion of the world's retailing industry, they can determine the measures while the manufacturers must follow. It may intuitively become an incentive to reduce the selling price to enhance market demand and profit [273]. Notably, it has been proven, in some cases, that retailer-led models lead to better results for the total CLSC with price and effort dependent demand [105] and also lead to a higher return rate, which is beneficial for decreasing the market price for the consumers [90].

In recent years, there has been a significant increase in the power of independent collectors taking a leadership role in the corresponding CLSCs. The prevalence of companies and plans such as Gem High-Tech, ReCellular, SIMS Metal Management, IBM's Global Asset Recovery Services, and AER Worldwide, reputed for e-waste, waste auto-collection, and resource recovery, are popular examples in the literature. However, the collector-led model is not always the most effective model for collecting used products [121]. Another

works considered a supplier who has more bargaining power than a retailer [111]. Due to the remanufacturing process's importance, giant companies (third parties) oblige other CLSC members to become their followers [58]. Some works showed that a balanced supply chain power relationship is the best for customers and the environment, and in contrast, as expected, an imbalanced supply chain power relationship is beneficial for the manufacturer and the retailer [15]. Furthermore, Mi, Huang, Wang, Tsai, Li and Wang [90] analyzed and found that when the transfer price is quite high, the scenario with no channel leader in the supply chain results in a higher return rate.

Practically, important parameters such as demand, price, and especially return rate are time dependent [27,203]. The current review shows that non-static models are necessary in some cases (for example, [95] when firms offer a variable rebate) and strategically are better (see [92] for the advantage of the dynamic pricing strategies with time-varying quality characterized by reference quality to a long-term and cooperative CLSC). Different types of games such as differential games or long-term evolutionary games [6,90–92] are powerful methods for studying the effect of dynamic behavior in CLSC, such as considering dynamic returns [56]. However, there are still gaps to integrate them into the CLSC [89,93].

## 6. Conclusions

Over time, CLSC models have shifted to management process and coordination, in which adopting GT models helps for more realistic decision-making by supply chain players [94]. There is a growing number of publications on CLSC utilizing GT for remanufacturing, reusing, repairing, recycling, and refurbishing decisions. In this paper, we analyzed the collaboration structures of CLSCs that are designed based on GT. The discussed structures were graphically reviewed.

Regarding collection, the manufacturer is the primary collector in the investigated models. A comparison of manufacturers' and retailers' performances on collection took precedence over the models that focus on different single channels. We found that the dual-channel collection method, in which two parties simultaneously collect the cores, is prioritized as compared with the other multiple-collection systems. Accordingly, dual collection by two retailers is the most studied case. Regarding hybrid collection approaches, the literature showed that retailer-manufacturer is superior to single-channel collection under the same conditions. A dual-selling channel managed by a manufacturer and remanufacturer is the case with the most attention for the future. In contrast to the dual collection cases, online selling received more attention.

In addition, different types of games (i.e., Stackelberg, Nash, Bertrand, Cournot, differential, game tree, repeated games, and evolutionary games) are categorized. Specifically, we determined that the Stackelberg case considering the manufacturer as the leader is mostly used to design a sequential game model. Nevertheless, recent years have seen a significant increase in the power of independent collectors taking a leadership role in the corresponding CLSCs.

According to our investigation, most of the studied models are steady-state models. The results reveal that 13 percent of the papers formulated CLSC models based on GT in the form of stochastic, fuzzy, or risk models. Among the three main reprocessing concepts, which are remanufacturing, recycling, and refurbishing, remanufacturing is the concept mostly taken into consideration in the reviewed CLSC models. Although other parties (i.e., suppliers, remanufacturers, and retailers) are also considered to be reprocessors individually, it is shown that single reprocessing by a manufacturer party took priority. In the dual reprocessing category, manufacturer-remanufacturer and two manufacturers are the most common ones in the literature.

In this review, there are some limitations. We only concentrated on WoS as the main database. Other databases, such as Scopus, could be used to find related papers. This review's scope was confined only to the models that formed a loop in one of the 3R forms. Therefore, game-based works merely studying the "green supply chain" were excluded. Only the loops considering EoU or end-of-life concepts were investigated. However, there

were models in which the loop is formed because of defective items. We did not consider such models. In addition, the period of the study was limited to a specific period.

## 7. Future Research Directions

There are still topics requiring further research. We suggest the following areas:

Only a few papers designed CLSCs with either multiple selling [4] or multiple collecting channels [53,157]. Particularly, when feeding different types of customers, each group may require its own dedicated collection-selling channel (tailor-made channel). An example from practice is collecting toner cartridges from individual households versus companies. Having non-homogenous multi-channels (e.g., brick vs. internet) changes the design of the CLSC. The assumption of having homogenous parties such as consumers is dominant when formulating CLSCs. However, relaxing this assumption changes the story (e.g., heterogeneous manufacturers). Therefore, there is still a shortcoming to deal with multiple CLSC channels. GT has great potential in this area.

With respect to the collection process, there are still some topics for further research. Although some companies employ an online method (e.g., check olx.in) to collect all types of used items such as mobile phones and electronic instruments, there is still a literature gap. Governments have taken responsibility for collecting cores in many countries, providing collection infrastructures such as accessibility of drop-off sites and pick-up services [158]. However, the contribution of the literature is limited to enforce a reward-penalty mechanism by the governments. Hence, the impact of a government's policies on the collection channel needs more investigations.

In the literature, cooperation and competition within a loop are well-discussed topics, and the discussions are even growing and improving. However, the connection of independent loops is still in its infancy [274]. Competition and cooperation of two independent CLSCs are depicted in Figure 8 as a novel idea. Therefore, according to GT and regarding the structure of the considered loops, different scenarios could be discussed with respect to centralized (cooperative) and decentralized (non-cooperative) or Nash game policies. Future works could study the price effect on the strategies across different loops and examine the impact of different parameters on optimal strategies.

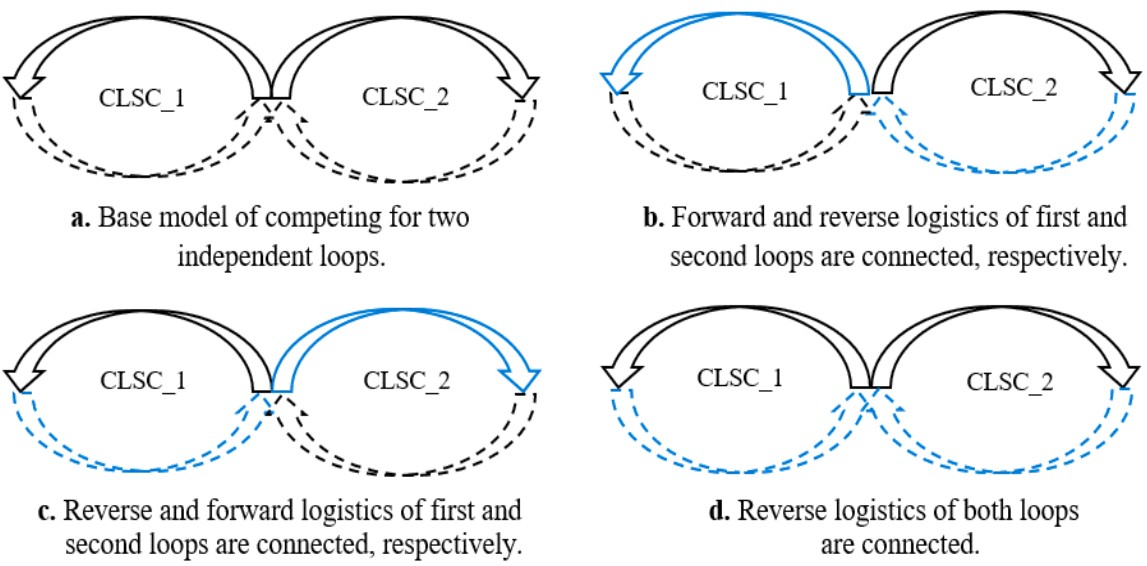

**Figure 8.** Sample relations of two competing CLSCs (Source, original by authors).

The CLSC, in reality, is a complex problem such that it can make it difficult to establish a long-term strategy. In order to resolve this issue, the evolutionary game is an efficient tool in the area of GT. However, it needs more investigations. Notably, the current works usually consider the CLSCs' performances from the perspective of two populations. Taking

more populations into account to interpret the behavior of players is a valuable direction for future research.

Unlike the CLSCs developed with only one period, one could consider extending to multiple periods in the future. Because of the dynamic nature of the process, such as consumer return behavior, a multi-period CLSC could be analyzed with a dynamic game to study the impact of possible nonstationary material flows on the channel choice decision. In some cases, it is an advantage and also a necessary (offering a variable rebate by a company, for example). A future novel idea to explore would be formulating a CLSC dependent on the processing time of remanufactured products as it is less than the processing time of new products.

There are not many models with more than two parties to do the reprocessing [53,76]. Multiple reprocessor parties can increase the performance of the CLSC. Moreover, multi-channel models that introduce different types of products such as new, remanufactured, and refurbished products simultaneously could enrich the literature on pricing decisions in the CLSC. This requires new models, including GT.

Stochastic processes in the CLSC should be further investigated, as most of the present models are deterministic. Parameters such as demand in the forward follow and return rate in the reverse direction are inherently stochastic.

Finally, an extension of this work would be to analyze the parameters and decision variables in the CLSCs as some of them, such as return rate and demand, have different forms in the reviewed papers. It would also be interesting to study the quality status of cores by reviewing quality functions [275].

Overall, this paper suggested a general framework for studying game theory's contribution to CLSC models. Similarly, this procedure can be applied to analyze the recently developed models [106,112,270,271,276] in this context.

**Supplementary Materials:** The following are available online at https://www.mdpi.com/2071-1050/13/3/1397/s1, Table S1: Categorization of the references regarding the structure, game, reprocessor party, and planning horizon.

**Author Contributions:** Conceptualization, E.S.; methodology, E.S. and S.D.F.; software, E.S.; validation, E.S. and S.D.F.; formal analysis, E.S.; investigation, E.S. and S.D.F.; resources, E.S.; data curation, E.S.; writing—original draft preparation, E.S.; writing—review and editing, E.S. and S.D.F.; visualization, E.S.; supervision, S.D.F.; project administration, E.S and S.D.F. All authors have read and agreed to the published version of the manuscript.

**Funding:** This research received no external funding.

**Institutional Review Board Statement:** Not applicable.

**Informed Consent Statement:** Not applicable.

**Data Availability Statement:** Not applicable.

**Acknowledgments:** We would like to thank Shahrzad Faghih-Roohi who checked the final format of the paper.

**Conflicts of Interest:** The authors declare no conflict of interest.

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
