# Peer review of "Analyzing the Structure of Closed-Loop Supply Chains: A Game Theory Perspective"

_sustainability, doi:10.3390/su13031397_

Round 1
Reviewer 1 Report
I have carefully reviewed the previous reviewer comments and concluded that authors did not address seriously the previous reviewer comments, which has been mainly emphasized on shorten the literature, converge on specific goals. Instead, I noticed, the authors of papers make arguments against the reviewer comments, these are not relevant, it might be possible for authors it does not address, however, the best way is to build your justification why not using these reviewer comments.
I remain wonder, why authors are not taking up reviewer comments and expand the methodological part, and the purpose of the review article is to provide some contributions from a broader field, not showing many studies ( not summarizing the articles, as this study included more than 200 articles). In these cases, a literature review provides the basis for building a new conceptual model or theory, and it can be valuable when aiming to map the development of a particular research field over time. As mentioned previously in reviewer comments, there are several existing guidelines for literature reviews. Depending on the methodology needed to achieve the purpose of the review, state clearly what type of literature review authors address in this study. I may suggest to authors, a systematic review aims to identify all empirical evidence that fits the pre-specified inclusion criteria to answer a particular research question or hypothesis. Following previous reviewer comments, short the literature, perform new key term searches, establish data validity and reliability. Even so, when I performed the existing key term search, which authors have used, I did not find the said number of articles, as it uses the month of April 2020. however, it provides many different articles because authors did not carefully describe in detailed inclusion nor exclusion criteria, which undermines the validity of the systematic literature review methodology. In the appendix, as the previous reviewer has suggested authors provide a summary of selected studies and their key findings. The authors have not addressed the reviewer's comments.
Author Response
We continued based on the editor's decision.
Reviewer 2 Report
The manuscript conducted a comprehensive review of the CLSC literature, from the lens of game theory.
The paper was noted to be generally well-written and mature (The positive impact of a potential round of critical, peer-review is evident). It does offer contributions to the CLSC literature and to the wider CE research streams, and is well within the aim and scope of the Sustainability journal.
I do however recommend minor changes prior to publication:
- Fig. 1 is hardly discussed within the text. Please explain this Figure in more detail, as this is a very important aspect in a Systematic Literature Review (SLR) paper.
- Methodological sources on SLR are highly recommended to be included (e.g. see Tranfield's paper on SLR methodology).
- Fig. 6 is a very long figure, it tampers the flow of the paper and it is advised to be revised. The authors are suggested to extract and only include the key points of this Figure, improve its quality, and include the full version in the form of an Appendix if required.
Author Response
Comment#1: The manuscript conducted a comprehensive review of the CLSC literature, from the lens of game theory. The paper was noted to be generally well-written and mature (The positive impact of a potential round of critical, peer-review is evident). It does offer contributions to the CLSC literature and to the wider CE research streams, and is well within the aim and scope of the Sustainability journal. I do however recommend minor changes prior to publication.
[Response]: Thanks for the positive comments. Now it is the fourth revision of the paper. The changes have appeared in purple color.
Comment#2: Fig. 1 is hardly discussed within the text. Please explain this Figure in more detail, as this is a very important aspect in a Systematic Literature Review (SLR) paper.
[Response]: Fig. 1 is revised, and moreover, the relevant explanations in section 2.1 were modified to be more connected to this figure to increase the readability and make the details more transparent.
Comment#3: Methodological sources on SLR are highly recommended to be included (e.g. see Tranfield's paper on SLR methodology).
[Response]: To enrich the content of the paper an excellent review paper by Tranfield was added to the methodology section.
Comment#4: Fig. 6 is a very long figure, it tampers the flow of the paper and it is advised to be revised. The authors are suggested to extract and only include the key points of this Figure, improve its quality, and include the full version in the form of an Appendix if required.
[Response]: Fig. 6 was totally revised which took a lot of time. We have increased the font size of the detail in all depicted structures to enhance the quality. As it is an important section and core of the research, we kept it in the current place. Especially there is a Table (Table 3) in the text, in section 3.5, along with explanations which are necessary to be close to Fig. 6.
Reviewer 3 Report
Dear Authors,
congratulation for your efforts. I would like to recommend a literature review studies to be referred to and in case add in the references if you want.
Try to see the methodological approach used in the selection and analysis.
Cheng W., et al. (2018). Green Public Procurement, Missing Concepts and Future Trends – A Critical Review, Journal of Cleaner Production, Available online 5 December 2017, ISSN 0959-6526, https://doi.org/10.1016/j.jclepro.2017.12.027
and also this one:
Testa F., Nucci B., et al. (2017). Removing obstacles to the implementation of LCA among SMEs: a collective strategy for the valorization of Recycled Cardato. Journal of Cleaner Production, 156, pp. 923-931, doi.org/10.1016/j.jclepro.2017.04.101
thank a lot
Author Response
congratulation for your efforts. I would like to recommend a literature review studies to be referred to and in case add in the references if you want. Try to see the methodological approach used in the selection and analysis.
Cheng W., et al. (2018). Green Public Procurement, Missing Concepts and Future Trends – A Critical Review, Journal of Cleaner Production, Available online 5 December 2017, ISSN 0959-6526, https://doi.org/10.1016/j.jclepro.2017.12.027
and also this one:
Testa F., Nucci B., et al. (2017). Removing obstacles to the implementation of LCA among SMEs: a collective strategy for the valorization of Recycled Cardato. Journal of Cleaner Production, 156, pp. 923-931, doi.org/10.1016/j.jclepro.2017.04.101
[Response]: Thanks for such a nice comment. The first paper is cited in the introduction section, line 89, as it has a similar method. The changes have appeared in purple color.
Reviewer 4 Report
Much of the contents provided in the introduction part should be moved into a [new] second section consisting of two sub-sections: 2.1. Background on GT, 2.2. A critical review of the existing review papers.
The introduction section should be re-structured to 4-5 paragraphs; it should begin with background on the importance of reverse logistics and closed-loop supply chain ideally with including some practical statistics highlighting the importance (paragraph 1); introduction should continue with an overview on the academic methods in general and GT in particular; the role of GT in the development of CLSC should be highlighted here in paragraph 2. Paragraph 3 should review THE MOST RELEVANT existing review papers with a critical lens and highlight the GAP. Paragraph 4 should clarify the contribution(s) of your work, i.e. the questions your research will answer. The section should be concluded by providing the outlines.
Considering the abundance of review papers for this particular topic, the stated academic gap should be convincing to me and your potential readers. In its current form, I do not see novelty and a need for a fresher review paper.
Additional bibliometric analysis may add value: for example, the most cited papers, scholars, and universities; as well as the countries of affiliation for providing insights into the nature of research in the developed and developing economies.
You need to ensure that the most recently published papers are cited, and ideally, included in your analysis. Following is one example:
Wei, Fangfang, et al. "Contract vs. Recruitment: Integrating an informal waste merchant to a formal collector for collection of municipal solid waste." Journal of Cleaner Production (2020): 125004.
The way figure 6 is organized is misleading and unclear. You need to make the figure self-sufficient; readers should be able to easily understand the meaning you want to convey. I also recommend to split it into different figures based on the categories you introduced.
You mentioned that 196 structures are identified; I believe these structures can be categorized using additional theoretical and practical lens to elevate the significance of your contribution.
I would like to see some practical implications in the discussion section. You also need to make a deeper connection between this research with the triple bottom line (social, economic, and environmental pillars of sustainability). In so doing, you may make two sub-sections: practical and academic implications.
Overall, there are many typos, grammar mistakes, and wrong wording throughout the paper that should be corrected before the next submission. For example "give" (line 13) is incorrectly used.
My major concern is the contribution of your study. I hope my suggestions can help improve the contribution of your work to an acceptable level.
Author Response
Comment#1: Much of the contents provided in the introduction part should be moved into a [new] second section consisting of two sub-sections: 2.1. Background on GT, 2.2. A critical review of the existing review papers.
The introduction section should be re-structured to 4-5 paragraphs; it should begin with background on the importance of reverse logistics and closed-loop supply chain ideally with including some practical statistics highlighting the importance (paragraph 1); introduction should continue with an overview on the academic methods in general and GT in particular; the role of GT in the development of CLSC should be highlighted here in paragraph 2. Paragraph 3 should review THE MOST RELEVANT existing review papers with a critical lens and highlight the GAP. Paragraph 4 should clarify the contribution(s) of your work, i.e. the questions your research will answer. The section should be concluded by providing the outlines.
[Response]: Thanks for your recommendation. The present format is based on the comments we received from previous rounds of revision. Actually, this is the fourth round of revision. Previously it had two sections. However, after receiving suggestions and comments from other reviewers, we changed it to the present format.
Comment#2: Considering the abundance of review papers for this particular topic, the stated academic gap should be convincing to me and your potential readers. In its current form, I do not see novelty and a need for a fresher review paper.
[Response]: We have explained the contribution of the paper in lines 129-150. This is the first time that structures of CLSCs are reviewed providing and concentrating detail regarding the game theory. If you are aware of any review paper, please let us know then we can add it to Table 1 and explain the differences as we did through the introductions section. Novelty and the differences of this review from the previous ones are explained in the introduction section in lines 107-128. You can also see the supplementary file. Please note we provided a framework to discuss CLSCs designed based on GT.
Comment#3: Additional bibliometric analysis may add value: for example, the most cited papers, scholars, and universities; as well as the countries of affiliation for providing insights into the nature of research in the developed and developing economies.
[Response]: This is not the concern of the paper. Please note it is not a bibliometric study. It is a content-based review paper. However, we have provided information regarding the year of publication and investigated journals.
Comment#4: You need to ensure that the most recently published papers are cited, and ideally, included in your analysis. Following is one example:
Wei, Fangfang, et al. "Contract vs. Recruitment: Integrating an informal waste merchant to a formal collector for collection of municipal solid waste." Journal of Cleaner Production (2020): 125004.
[Response]: The study period is included in the paper, and the limitation of the research is discussed. This is a growing area, and for sure, it is not possible to include all papers. That is why each review paper, and similarly, our review considers a time horizon. The mentioned article has been online on 16 November 2020 and still in press and does not consider the forward direction (i.e., selling new products), which means it is out of the scope of this paper. Please see the structures that we depicted. Meanwhile, if you want to see this article, please let us know where we should cite it. Then we can cite your article.
Comment#5: The way figure 6 is organized is misleading and unclear. You need to make the figure self-sufficient; readers should be able to easily understand the meaning you want to convey. I also recommend to split it into different figures based on the categories you introduced. You mentioned that 196 structures are identified; I believe these structures can be categorized using additional theoretical and practical lens to elevate the significance of your contribution.
[Response]: Thanks for your suggestions. Please read the figures based on Table 3 and explanations in section 3.5. There were minor errors that were revised. Please refer to the supplementary file and table in the appendix. Regarding the splitting, please see the last column in Table A in the appendix.
Comment#7: I would like to see some practical implications in the discussion section. You also need to make a deeper connection between this research with the triple bottom line (social, economic, and environmental pillars of sustainability). In so doing, you may make two sub-sections: practical and academic implications.
[Response]: As it was explained, the present structure is the result of the previous round of revisions. We did not discuss the TPL in this review paper. The target is providing a framework to discuss the CLSC that are designed based on the GT.
Comment#7: Overall, there are many typos, grammar mistakes, and wrong wording throughout the paper that should be corrected before the next submission. For example "give" (line 13) is incorrectly used.
[Response]: A native editor has improved the paper.
Comment#8: My major concern is the contribution of your study. I hope my suggestions can help improve the contribution of your work to an acceptable level.
[Response]: Please see the response to comment 2. If you think any relevant paper during the considered period is missed, you can offer.
Round 2
Reviewer 4 Report
Except for comments 4 (I am convinced that the reference does not fall into the scope of the manuscript in terms of structure) and 7(proofreading), none of the other comments are addressed.
This work, if accepted, will be published in a 2021 issue of the journal, hence, it is not acceptable to review the papers published until 2019. A fresh review of 2020 papers is missing, and this aspect may highlight your contribution given that there are older reviews around this topic. The practical and academic implications section should also be considered to add value to your findings.
I would like to ask the authors to revisit the comments I provided in the first round of review.
Author Response
First of all, thanks a lot for your time to review our paper and your useful comments. Here our response to your comments (italic). The changes are shown with the red color through the revised version.
Except for comments 4 (I am convinced that the reference does not fall into the scope of the manuscript in terms of structure) and 7(proofreading), none of the other comments are addressed. I would like to ask the authors to revisit the comments I provided in the first round of review.
Response: We revised the paper again regarding the previous comments as explained below. We tried to follow your comments.
This work, if accepted, will be published in a 2021 issue of the journal, hence, it is not acceptable to review the papers published until 2019. A fresh review of 2020 papers is missing, and this aspect may highlight your contribution given that there are older reviews around this topic. The practical and academic implications section should also be considered to add value to your findings.
Response: Thanks for your recommendation. We have added important papers from 2020 into the last section to make the references fresh and update in this context. In this way, we also cited your previous work titled “Contract vs. recruitment: Integrating an informal waste merchant to a formal collector for collection of municipal solid waste.” We apologize that we ignored it in the previous revision. Besides, the discussion section is re-organized to the practical and academic sections.
Your previous comments as below:
Comment#1: Much of the contents provided in the introduction part should be moved into a [new] second section consisting of two sub-sections: 2.1. Background on GT, 2.2. A critical review of the existing review papers.
The introduction section should be re-structured to 4-5 paragraphs; it should begin with background on the importance of reverse logistics and closed-loop supply chain ideally with including some practical statistics highlighting the importance (paragraph 1); introduction should continue with an overview on the academic methods in general and GT in particular; the role of GT in the development of CLSC should be highlighted here in paragraph 2. Paragraph 3 should review THE MOST RELEVANT existing review papers with a critical lens and highlight the GAP. Paragraph 4 should clarify the contribution(s) of your work, i.e. the questions your research will answer. The section should be concluded by providing the outlines.
[Response]: Thanks for your comments. We have revised the introduction section based on your suggestions. We also considered section 2 based on your recommendation.
Comment#2: Considering the abundance of review papers for this particular topic, the stated academic gap should be convincing to me and your potential readers. In its current form, I do not see novelty and a need for a fresher review paper.
[Response]: We hope that the provided explanations and changes based on your suggestions in previous comments clarify our review's contributions in the present format. In the present format, paragraphs 3 and 4 show the differences between our work with the recent review papers.
Comment#3: Additional bibliometric analysis may add value: for example, the most cited papers, scholars, and universities; as well as the countries of affiliation for providing insights into the nature of research in the developed and developing economies.
[Response]: We have mentioned in the revised format, in section 2.1, that “The first seminal paper discussing collection channels and designing a game on a CLSC is the one by Savaskan, Bhattacharya [9], which is the most cited paper in this field.” We also added another figure regarding the contribution of different countries based on the published papers to improve the section 3.
Comment#5: The way figure 6 is organized is misleading and unclear. You need to make the figure self-sufficient; readers should be able to easily understand the meaning you want to convey. I also recommend to split it into different figures based on the categories you introduced. You mentioned that 196 structures are identified; I believe these structures can be categorized using additional theoretical and practical lens to elevate the significance of your contribution.
[Response]: Thanks for your suggestions. The depicted structures reflect the complexity of models regarding applying one of the categorized factors and reprocesses. It is possible to check which concepts are appeared in a model simultaneously as a measure of complexity.
For example, structure 1 shows that there is a manufacturer (M) who sells the new products in the forward channel (blue arrow) to a retailer (R). In the reverse channel (green arrow), the manufacturer collects the used products from the market (D) and implements the remanufacturing process (it is shown as *). The manufacture plays as the leader of the game (it is shown as the red color). The definition of applied notations, arrows, and colors in the CLSCs structure are presented in Table 3 and through the text in section 4.5. As it is explained, the reader can refer to the last column called "structure" in Table 1A (in Appendix) to know which paper used this structure. For instance, structure 1 is used in Savaskan, Bhattacharya [9], and Atasu, Toktay [59], among others.
Please note splitting the figures in this way is not possible. Because there are main common features between structures, however, if you or the editor have a way, please suggest it in detail.
Comment#6: I would like to see some practical implications in the discussion section. You also need to make a deeper connection between this research with the triple bottom line (social, economic, and environmental pillars of sustainability). In so doing, you may make two sub-sections: practical and academic implications.
[Response]: Thanks for your good suggestion. The discussion section is revised according to your proposal.
Once again thanks a lot for your constructive comment.
Round 3
Reviewer 4 Report
Much of my comments are well addressed. However, the authors should include 2020 published works within the content analysis. On this basis, the information on Tables 1-2 as well as line 171 should be updated. I hope the authors understand my concern and address it carefully. Besides, the title of section 2 should be changed to relevant works.
Author Response
Comments: Much of my comments are well addressed. However, the authors should include 2020 published works within the content analysis. On this basis, the information on Tables 1-2 as well as line 171 should be updated. I hope the authors understand my concern and address it carefully. Besides, the title of section 2 should be changed to relevant works.
Response: Thanks for your positive feedback. The content analysis is updated regarding the added references, and accordingly, the information in Tables 1-2 and line 171 are updated based on your concern. Also, the title of section 2 is changed.
We appreciate your careful consideration in reviewing the paper.
This manuscript is a resubmission of an earlier submission. The following is a list of the peer review reports and author responses from that submission.
Round 1
Reviewer 1 Report
The topic of the paper is potentially interesting. However, I think there are some critical points and weaknesses that impede to publish the article
-P.1, Line 19, i suggest authors to highlight the governance mechanis used in supply chain., such as, contract and relational and its importance.
- P.1, Line 23-30, I suggest authros highlighs topic importance in relation to the suggestion articles. https://doi.org/10.3390/su12072993
- there is a lack of definition for some of the main concepts: Due to the lack of clear definition about the main concepts. In the introduction, you need to connect the state of the art to your paper goals and highlight what gaps has been left in the previous literatur reviw studies. Also clealry mention study contribution. Aim of the study need to redefine.. Please follow the literature review by a clear and concise state of the art analysis
1- Why authors choose “closed-loop supply chain” to locate articles. Also elaborate inclusion and exclusion criteria with more detials. Compare WoS dagtabase with other, why not authorselect scopus? Provide some disucssion
-3.1 mateiral selection, provide literature on systemati literature review and its important as a research method.
-Mvoe table Table 4. To appendix.
Review analysis is very long, and it seems ambiguous. It can not get reader attention. I suggest authors not to put more focus on writing literature, instead, support your review results with the previously established literature and give your arguments. This should be particularly done to ensure the appropriateness and the robustness test for the effectiveness of the results. Results and interpretations: the structure of the results and discussion is not really logical, which results in several redundancies and in a text going several times back and forth between topics. Most interpretations are correct, but some must be questioned. https://doi.org/10.1007/978-3-319-74225-0_9
- The discussion is relatively simple and insufficient. I recommend strengthening the comparison with previous research.
- The expected contribution of the current research will be limited based on the current version. If the authors could provide good theorization in this study, which might help open some new insights on the contribution of this research.
Reviewer 2 Report
The manuscript conducted an extensive review into the works that studied Game Theory in the context of Closed Loop Supply Chain Management for a step towards Circular Economy. A current map of the extant literature has been drawn, key themes obtained and future research directions outlined.
The research area is of emerging nature, and is well within the aim, objectives and scope of the Sustainability journal. On the other hand, the following points are required to be taken into account prior to publication:
- CLSCs are not a tool yet, probably a concept or an approach.
- The inclusion of the key aspects of the research method adopted and provision of the key findings of the study are recommended in the Abstract section.
- Prior to discussion of the links between CLSC and Game Theory, an introduction into the GT would add value to the first section of the manuscript, especially for the wider reader base.
- The central argument of the research with ref. facilitating CLSC through GT can be fortified further in the first and second sections.
- The research methodology adopted needs to be justified more carefully. Simply referencing a previously published paper is not enough. Please note that this work does make tons of contributions and may have inspired or may have acted as a guiding light for the authors, but it is not a methodological source. I suggest the authors check methodological papers on literature reviews, such as Tranfield’s paper.
- Did the authors conduct a systematic literature review? If yes, why?
- What analysis method(s) did the authors utilise while analysing the many papers that they have identified? How did they reach the results? Details of the analysis methods are required to be included in the methodology sections. It seems like content analysis was adopted, but why? What other methods were available? How was this method exactly adopted? And many more questions could be asked here…
- The formatting of tables 2 and 3 need to be improved. They are difficult to flow in their current format.
- A descriptive and visual summary of the key findings in Table 4 would highly help the flow of the paper. In fact, the table 4 could be provided in the form of an Appendix and a summary figure could replace this massive table in the main text. The flow of the paper is tampered by this very long Table currently.
- I suggest revising the name of Section 6 as “Discussion”.
- The limitations of the research are required to be discussed.
- The last section can be divided into two sections as “Conclusions” and “Future Research Directions”. The conclusions sections will then provide an overview of the key findings of the study and their limitations. The manuscript can then finish off with the recommendations for future research.
Round 2
Reviewer 1 Report
Author has addressed all the reviewer comments, but structure of the manuscript has been changed much and it contains a lot of text. The topic is very relevant and the manuscript reads very well, and interesting too. However, and based on stated manuscript criteria for journals review selection, it is unclear why a number of highly relevant journals are not included in this review study.
Although authors have established a good theoretical gap Lines 69-72 while discussing is not clear and author need to establish a clear research gap, I suggest establish research gap with appropriate systematic literature review studies. What literature review studies have been conducted previously in the domain of the CLSC and why this literature review is important.
-However, the introduction should have 1) a concise but full justification of the topic's importance both academically and practically, and 2) an explanation of the gaps both in research and practice. Please review appropriate literature in the introduction, with the research question clearly arising from that review. Also, line 78-87, these lines should move under the solution methodology.
- Line 73, the study objective is not clear and why author(s) select such a large sample size , that is, 220 articles. I have some concern on SLR validity. Need epxlaination.
-Line 131, I have serious concern here, it seems that author reach directly to the study contribution without appropriate literature references in the previous discussion. I did not find these lines of arguments are justified
-Line 146-, I am wonder how author choose the key words only “closed-loop supply chain” form 2004 and 2019.I am wonder why authors are failed to describe the detailed inclusion and exclusion criteria.
-Major areas of agreement and disagreement in the literature should be discussed. The discussion should tie the study into the current body of literature, provide its significance, and make logical interpretations from the literature review
- It would be best to indicate in some way the selected articles in the literature review. (I do not know clearly whether the reference set contains the all selected literature review article or not.) In some way, the selected literature review should be listed (whether as an appendix or by using a symbol such as an asterisk in the reference set). There should be a separate/special reference set for publications cited in the text. (There may be some duplication, but the author can figure out some reasonable system.)
- Please explain clearly what the contributions to theory and practice are, because it is confusing. Moreover, one of the most important contributions in a literature review is to provide opportunities for future research, a more explaination is needed.
Reviewer 2 Report
The authors provided full and credible responses to all points raised, developing their manuscript in the light of the reviewer feedback.
Round 3
Reviewer 1 Report
The topic is very relevant, and the manuscript reads very well, and interesting too. I have already provided my detailed comments earlier. However, the present manuscript could be improved. For example, the authors can specify primary screening databases. It is not clear what databases the authors have used for their searches. why WoS data is selected, my serious concern about the validity and reliability of the data. The key term search criteria are not convincing. When I performed this key term search and I found different results because the authors did not clearly provide inclusion or exclusion criteria. This undermines the reliability of the key term search. To be clear, we know there is no way to include all (relevant)journals in a single review study, but there must be a clearly stated selection criterion that categorically explains inclusion and/or exclusion. Locating studies: Please provide a table showing these keywords combinations. Please justify why you choose this engine and make clear what kind of document was considered (article, review, conference paper, book, etc.).However, in my opinion, the author needs to convince readers that the search terms themselves have not to affect reliability. The purpose of the literature review is to provide concentrated literature to the reader, but the author mainly emphasizes explaining the literature review. I suggest the authors remove some extra text from the manuscript